# Phytochemical and pharmacoinformatics analysis of a traditional antipsoriatic oil formulation for its potential against proinflammatory cytokines TNF-α and IL-17A

Nayak Deeksha Dayanand[1], Vasudha Devi[1,2]*, Shama Prasada Kabbekodu[3], Arul Amuthan [4,5], Sathish Pai B[6], Rajasekhar Chinta [7]*

1 Department of Pharmacology, Kasturba Medical College, Manipal, Manipal Academy of Higher Education, Manipal, Karnataka, India, 2 Centre for Cardiovascular Pharmacology (CCP), Manipal Academy of Higher Education, Manipal, Karnataka, India, 3 Department of Cell and Molecular Biology, Manipal School of Life Sciences, Manipal Academy of Higher Education, Manipal, Karnataka, India, 4 Division of Pharmacology, Department of Basic Medical Sciences, Manipal Academy of Higher Education, Manipal, Karnataka, India, 5 Coordinator, Division of Siddha, Centre for Integrative Medicine and Research (CIMR), Manipal Academy of Higher Education, Manipal, Karnataka, India, 6 Department of Dermatology, Venereology and Leprosy, Kasturba Medical College, Manipal, Manipal Academy of Higher Education, Manipal, Karnataka, India, 7 Department of Pharmacology, Manipal University College Malaysia, Melaka, Malaysia

* diya.chinta@manipal.edu.my (RC); vasudha.devi@manipal.edu (VD)

## Abstract

Psoriasis, a persistent inflammatory condition with a complex origin, lacks a definitive cure despite the availability of diverse treatments. Vetpalai thailam (VT), a Siddha formulation, is commonly used for addressing various skin ailments. Analysis via GC-MS of a sample of VT oil unveiled the existence of 65 compounds. These identified phytochemical structures were then examined through molecular docking against the proinflammatory cytokines TNF-α and IL-17A, resulting in the shortlisting of seven compounds. Among these, four displayed binding potential against TNF-α, while three exhibited similar potential against IL-17A. Further, the individual shortlisted phytochemical and protein complexes were subjected to molecular dynamics and documented their RMSDs, RMSFs, SASA, Rg, MMPBSA, and PCA profiles. Other informatics web tools were employed to predict physicochemical properties, bioactivity, drug-likeness, and toxicity scores. Three compounds identified from VT oil displayed binding potential against IL-17A, showcasing binding energies varied between −8.427 and −6.739 kcal/mol, while four compounds exhibited potential against TNF-α, with binding energies ranging from −9.873 to −8.644 kcal/mol. The physicochemical attributes, bioactivity, compliance with Lipinski's rule of five, and ADMET profiles of the shortlisted compounds demonstrated favorable pharmacokinetic characteristics. Consequently, this research provides valuable insights into the binding ability of phytoconstituents of VT oil against IL-17A and TNF-α, paving the way for the development of novel drugs for the treatment of psoriasis.

**Data availability statement:** All files are available from the Zenodo database: 10.5281/zenodo.15925012.

**Funding:** The author(s) received no specific funding for this work.

**Competing interests:** The authors have declared that no competing interests exist.

## Introduction

Psoriasis is a chronic autoimmune and inflammatory skin disorder that affects approximately 1–3% of the global population, influenced by both genetic and environmental factors [1]. Among numerous cytokines involved in the pathogenesis of psoriasis, TNF-α and IL-17A are particularly significant due to their synergistic effects in amplifying chronic inflammation. TNF-α, primarily secreted by macrophages and T cells, activates key signaling pathways such as NF-κB and MAPKs, which in turn stimulate the expression of downstream inflammatory mediators [2]. When acting together, TNF-α and IL-17A elicit a much stronger inflammatory response than either cytokine alone, leading to excessive keratinocyte proliferation and further cytokine release [2]. In psoriatic lesions, Th17 cells secrete cytokines such as IL-17A, IL-12, IL-22, and IL-9, which further stimulate keratinocytes. This stimulation leads to abnormal keratinocyte proliferation and the release of additional inflammatory mediators like IL-6, IL-8, TNF-α, CCL20, and CXCL1/2/3/5/8 in a loop that contributes to the progression of the disease [3]. IL-17A, predominantly produced by Th17 cells, acts on epithelial and stromal cells to induce chemokine and antimicrobial peptide production, promoting neutrophil recruitment and tissue inflammation [4]. The complementary roles of TNF-α and IL-17A in driving inflammation make them critical therapeutic targets and biomarkers in psoriasis research. Inhibiting these cytokines has been shown to induce broad molecular changes in psoriatic lesions, affecting immune responses, tissue remodeling, and metabolic activity [4].

Multiple biologic agents were developed targeting these proinflammatory cytokines. Antibodies such as secukinumab, ixekizumab, and brodalumab have been designed to inhibit IL-17A signaling and are currently used in the treatment of psoriasis [3]. Similarly, elevated levels of TNF-α in both blood and lesional skin of individuals with active disease have led to the use of TNF-α inhibitors (adalimumab, infliximab), which exert their therapeutic effects partly by suppressing Th17 cell activity [5]. Although these biologic agents are highly effective, their use is often restricted due to concerns about toxicity and high cost [6–9]. In the management of psoriasis, topical therapies are typically the first-line approach for mild and localized cases and are often combined with systemic treatments or phototherapy to enhance efficacy and reduce long-term progression [10]. Methotrexate is a commonly used topical agent, but its application is limited by adverse effects such as skin irritation, hives, eczema, inconsistent absorption, and the cost [11]. Current clinical guidelines continue to recommend topical corticosteroids, vitamin D analogues, and calcineurin inhibitors as the primary first-line therapies for managing mild psoriasis and to limit the disease progression but the adverse effects of them are still a major concern [12,13]. Despite the availability of various systemic, topical, and biologic treatments, a definitive cure for psoriasis remains elusive [14].

Traditional and folk medicines have gained renewed interest in managing various diseases, as they are typically derived from natural sources and are more affordable [15]. Vetpalai thailam (VT) is the mainstay of the traditional Siddha way of treatment for psoriasis. It is prepared by soaking 1 kg of fresh *Wrightia tinctoria* leaves in 1 liter of coconut oil, this is further filtered, and sun heated. This oil is extensively

prescribed as a topical application in traditional Siddha Medicine [16–18]. *Wrightia tinctoria* was studied and reported its anti-inflammatory activity evidenced by significantly decreased carrageenan induced rat paw oedema and reduced weight of the cotton pellet granuloma [19]. This preparation has shown significant anti-inflammatory activity in animal models [20]. A 90-day repeated dose toxicity study on rats assured the safety of using VT oil as no signs of toxicity and behavioral changes were observed [21]. Current research focuses on the phytochemical and pharmacoinformatics analysis of VT oil preparation to elucidate its potential against key inflammatory proteins (TNF-α, IL-17A) involved in psoriasis.

## Materials and methods

### GCMS analysis

VT was obtained from Sivasakthi Pharmaceutical Pvt Ltd, a company in Coimbatore, Tamil Nadu, India, holding GMP and ISO certifications. The Gas Chromatography-Mass Spectrometry (GC-MS) analysis took place at Analytical Research & Metallurgical Laboratories Pvt. Ltd. (ARML) in Bengaluru, Karnataka, India. To separate the active components of VT oil, it was dissolved in methanol at a concentration of 1 mg/mL. The GC-MS analysis utilized a Shimadzu GCMS-QP2010S instrument equipped with an RTX-5 column (30m length, 0.25 mm internal diameter, and 0.25 μm film thickness). The ion source temperature and interface temperature were set at 200°C and 280°C, respectively, with helium serving as the carrier gas at a flow rate of 1mL/min. The analysis involved injecting a 1 μL sample, and the compound identification was performed using GC-MS with spectral matching against the NIST05 and NIST05s libraries. Each identified compound entry included the hit number, library entry ID, library name (e.g., NIST05.LIB), similarity index (SI), molecular formula, CAS number, retention index, and compound name (CompName). (S1 File)

### Preparation of TNF- α and IL-17A structures for docking

The 3D X-ray crystallographic structures of IL-17A (PDB ID: 5HI5) and TNF- α (PDB ID: 2AZ5) were sourced from the Protein Data Bank (RCSB) in PDB format. Post-retrieval, the protein structures were refined by eliminating HETATOM coordinates associated with the bound ligands. The refined protein files were then viewed using the Swiss Protein Data Bank Viewer (SPDBV), conducted energy minimization, and the resulting optimized files were employed for molecular docking studies.

**Ligand preparation.** The 2D configurations of the plant components found in VT oil were retrieved from the PubChem Compound database. A total of 65 compounds were drawn, handled, and stored in MDL mol form with the aid of ChemSketch 2019 2.2 software by ACD Laboratories. Following this, Open Babel GUI (version 3.1.1) was employed to transform these configurations into PDB format. The ensemble of conformers for each compound was verified using the open source ENTOS ENVISION tool followed by the lowest energy conformers for each phytochemical was selected for molecular docking. Subsequently, the 2D layouts were converted to PDBQT with MGL tools and utilized for docking purposes.

**Molecular docking.** The phytochemicals identified from VT oil through GCMS analysis were subjected to molecular docking against two pivotal cytokines involved in the psoriatic inflammatory pathway, namely IL-17A and TNF-α, utilizing AutoDock Vina software from the Scripps Research Institute in the USA. The protein structures were prepared in MGL tools by incorporating polar hydrogen residues and Kollman atomic charges, assigning radii, and subsequently saving the file as PDBQT. The structures of phytoconstituents were configured to retain rotatable bonds. The docking calculations were configured for fixed protein and flexible ligand interactions using a sophisticated gradient optimization method in its local optimization [22]. Grid boxes were positioned at the binding sites of TNF-α and IL-17A similar to that of respective native ligands, with coordinates set at X: 107.640, Y: −29.892, Z: 1.991 Å for TNF-α, and X: 102.640, Y: 9.850, Z: 29.741 Å for IL-17A, each with a grid size of 40. The Vina application was executed through the command prompt (cmd), and from ten runs, the conformation with the lowest docked energy was scrutinized for each phytoconstituent. The final affinity

value was determined based on the average affinity of the best poses. The interactions of phytochemicals with respective proteins, protein-ligand conformations, non-polar interactions, and their interacting residues, and distances were comprehensively examined using the BIOVIA Discovery Studio Visualizer. To validate the docking process, the redocking method was employed utilizing the native ligand obtained from the co-crystallized structure and apo form of TNF-α. A comparison was made between the interactions of the native protein-ligand complex and the redocked complex, and the Root Mean Square Deviation (RMSD) of both ligand orientations within the protein binding site was recorded as < 2 Å (S1–S4 Figs, See: S2 File).

## Molecular dynamics simulations and binding free energy calculations

To investigate the dynamic characteristics of the selected phytoconstituents against TNF-α and IL-17A proteins, molecular dynamics (MD) simulations [23] were performed on the shortlisted docked complexes [24]. Four phytochemicals were individually complexed with TNF-α (PDB ID: 2AZ5), while three were complexed with IL-17A (PDB ID: 5IH5), and the apo forms of both proteins were also analyzed. The simulations were carried out using the Gromacs 2020.2 software package with the CHARMM-36m force field. The force field parameters for the phytochemicals were generated using the CGenFF program. The systems were neutralized with sodium and chloride ions and simulated at a temperature of 300 K and a pressure of 1 bar. The simulations underwent energy minimization, equilibration, and a 100 ns production run. Trajectory analyses [25] were performed to study RMSD, root mean square fluctuations (RMSF), radius of gyration (Rg), and hydrogen bond occupancy. The binding free energy of the system was calculated by using molecular mechanics-based scoring methods MM/PBSA. The calculations were based on a total of 300 snapshots of the complex, taken at 2 ps (picosecond) intervals from the last 2 ns (nanosecond) stable MD trajectories. The binding free energy was determined as the difference between the total free energy ($\Delta G_{com}$) of the ligand-receptor complex and the sum of free energy of individual receptors ($\Delta G_{pro}$) and ligand ($\Delta G_{lig}$) using the equation provided below:

$$\Delta G_{bind} = \Delta H - T\Delta S = \Delta G_{com} - [\Delta G_{pro} + \Delta G_{lig}]$$

The equation for calculating ΔG for the complex, receptor, and ligand is as follows:

$$\Delta G = \Delta E_{MM} + \Delta G_{sol} - T\Delta S$$

$\Delta E_{MM}$ = Molecular Mechanics Energy.
$\Delta G_{sol}$ = Solvation-Free Energy.
$T\Delta S$ = Entropy at given Temperature [26].

## Principal component (PCA) and eigenvector index analyses

We performed principal component analysis (PCA) to examine the conformational flexibility and structural variations of the system throughout a 100 ns molecular dynamics (MD) simulation. By evaluating the stable trajectory derived from the simulation, PCA enabled us to identify the key modes of motion that influence the system's overall dynamics. The primary collective motions were predominantly captured by the first 50 eigenvectors, also known as principal components (PCs), which represent the most significant conformational fluctuations. Among these, we focused on analysing the first two eigenvectors/PCs in detail, as they correspond to the most pronounced large-scale movements shaping the system's conformational landscape.

## ADMET analysis

Pharmacokinetic characteristics of the shortlisted ligands were assessed through absorption, distribution, metabolism, excretion, and toxicity (ADMET) prediction utilizing SwissADME and ADMETlab2.0 web servers. Parameters such as

molecular weight, number of rotatable bonds, H-bond acceptors, H-bond donors, Topological polar surface area (TPSA), adherence to Lipinski's rule, drug-likeness scores, and bioactivity prediction scores were determined. These data were then analyzed comprehensively to understand the pharmacological behavior of each compound.

## Results

### Phytochemical profiling of VT oil

Comprehensive phytochemical characterization of VT oil was performed using gas chromatography–mass spectrometry (GC-MS), which led to the identification of 65 distinct compounds. These constituents are detailed in S1 Table, See: S3 File, which includes each compound's retention time (RT), percentage area, molecular weight, molecular formula, and CAS registry number. The compounds are arranged in descending order based on percentage area, providing a clear representation of their relative abundance in the formulation. This ranking offers insight into the dominant chemical constituents likely contributing to the biological activity of VT. The Similarity Index (SI) scores ranged from 58 to 95, reflecting varying degrees of match confidence. To evaluate the reliability of compound assignments, the following similarity thresholds were used: SI > 90 was classified as very high confidence, SI 86–90 a**s** high confidence, SI 81–85 a**s** moderate confidence, SI 76–80 a**s** low confidence, and SI ≤ 75 a**s** very low confidence. While identifications with SI ≥ 85 are generally considered robust.

### Analysis of the interactions between short-listed ligands of VT oil and the binding site of IL-17A

Molecular docking was conducted to identify the top-performing compounds for IL-17A based on their minimal binding energy and number of interactions, as presented in Table 1. Within the crystal structure of IL-17A (PDB ID: 5HI5), the co-crystallized ligand (45, 20R)-7-chloro-N-methyl-4-f[(1-methyl-1H-pyrazol-5-y|)carbonyl]amino)-3, 18-dioxo-2, 19-diazatetracyclo[20.2.2.16.10.1~11, 15-]octacosa-1(24), 6(28), 7, 9, 11(27), 12, 14, 22, 25-nonaene-20-carboxamide) predominantly interacted with Leu97 and Pro63 residues, with fewer interactions with Trp67 of IL-17A dimer interface (S5 Fig, See: S2 File). Using this ligand as a reference, we analyzed the interactions of phytochemicals from VT oil against IL-17A. The shortlisted ligands (S6 Fig, See: S2 File) exhibited strong interactions with residues akin to the co-crystallized ligand, notably Leu97 and Pro63 (Fig 1). Monolaurin demonstrated significant interactions with the IL-17A binding surface, forming 11 interactions, including five hydrogen bonds with Leu97, two hydrophobic interactions with Pro63, and one with Leu97, with a minimum binding energy of −8.42 kcal/mol. This compound may be the most prominent among the three shortlisted candidates. The 2,3-bis(trimethylsilyloxy)propyl (9E,12E)-octadeca-9,12-dienoate (101278611) formed a total of 10 interactions, comprising three hydrogen interactions with Leu97, two hydrophobic interactions with Pro63, and two

**Table 1. Shortlisted phytoconstituents of VT and their interactions with IL-17A.**

| Sl. No | Phytoconstituents with structure | BE (kcal/mol) | No of interactions | Hydrogen interactions | Hydrophobic interactions |
|---|---|---|---|---|---|
| 1 | Monolaurin ($C_{15}H_{30}O_4$) | −8.427 | 11 | Leu 97 (5) | Pro 63 (2)<br>Val 98<br>Leu 97<br>Leu 99<br>Ile 96 |
| 2 | Trimethylsilyl 2,4-bis (trimethylsilyloxy) benzoate ($C_{16}H_{30}O_4Si_3$) (517875) | −7.839 | 4 | Ile 96<br>Leu 97 | Ile 96 (2) |
| 3 | 2,3 bis (trimethylsilyloxy) propyl (9E,12E)-octadeca-9,12-dienoate ($C_{27}H_{54}O_4Si_2$) (101278611) | −6.739 | 10 | Leu 97 (3) | Pro 63 (2)<br>Leu 97 (2)<br>Tyr 62 (2)<br>Leu 112 |

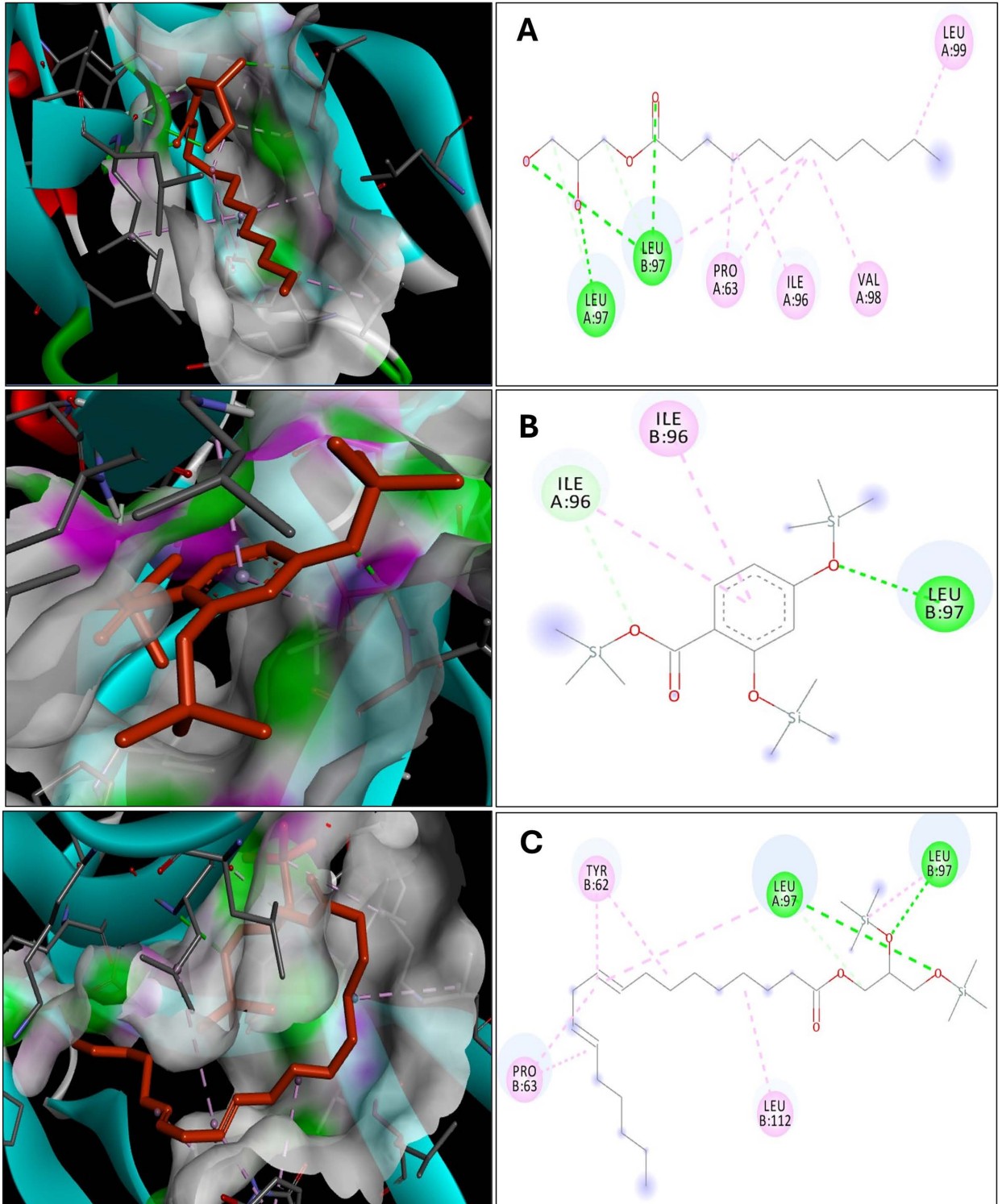

**Fig 1. Display of docking interactions made by the shortlisted phytoconstituents of VT with the binding site of IL-17A.** Light and dark green lines: hydrogen interactions (< 5 Å distance); purple & pink lines: hydrophobic interactions. A) Monolaurin B) Trimethylsilyl 2,4-bis(trimethylsilyloxy)benzoate (517875) C) 2,3 bis(trimethylsilyloxy)propyl (9E,12E)-octadeca-9,12-dienoate (101278611).

with Leu97, with a binding energy of −6.73 kcal/mol. Meanwhile, Trimethylsilyl 2,4-bis(trimethylsilyloxy)benzoate (517875), with a minimum binding energy of −7.83 kcal/mol, exhibited two hydrogen contacts, one with Leu97 and another with Ile96, alongside two hydrophobic interactions with Ile96 (Table 1).

## Dynamic behavior of IL-17A complexed with respective shortlisted ligands of VT oil during simulation

The RMSD analysis of IL-17A complexes of shortlisted phytochemicals (monolaurin, 517875) 101278611), revealed diverse patterns. Monolaurin displayed consistent RMSD values comparable to the apo form of IL-17A, while 517875 exhibited fluctuations throughout the simulation, and 101278611 displayed fewer fluctuations, remaining closer to the RMSD of the apo form of IL-17A (Fig 2a). Greater solvent-accessible surface area (SASA) trends were observed for IL-17A complexes with compound 517875 and compound 101278611 compared to monolaurin (Fig 2b). The radius of gyration (Rg) profiles depicted constant fluctuations for monolaurin and compound 517875 complexes, whereas compound 101278611 exhibited a more pronounced Rg trend throughout the simulation (Fig 2c). Hydrogen bond analysis revealed that monolaurin formed numerous hydrogen bonds, while the other two compounds interacted with a single

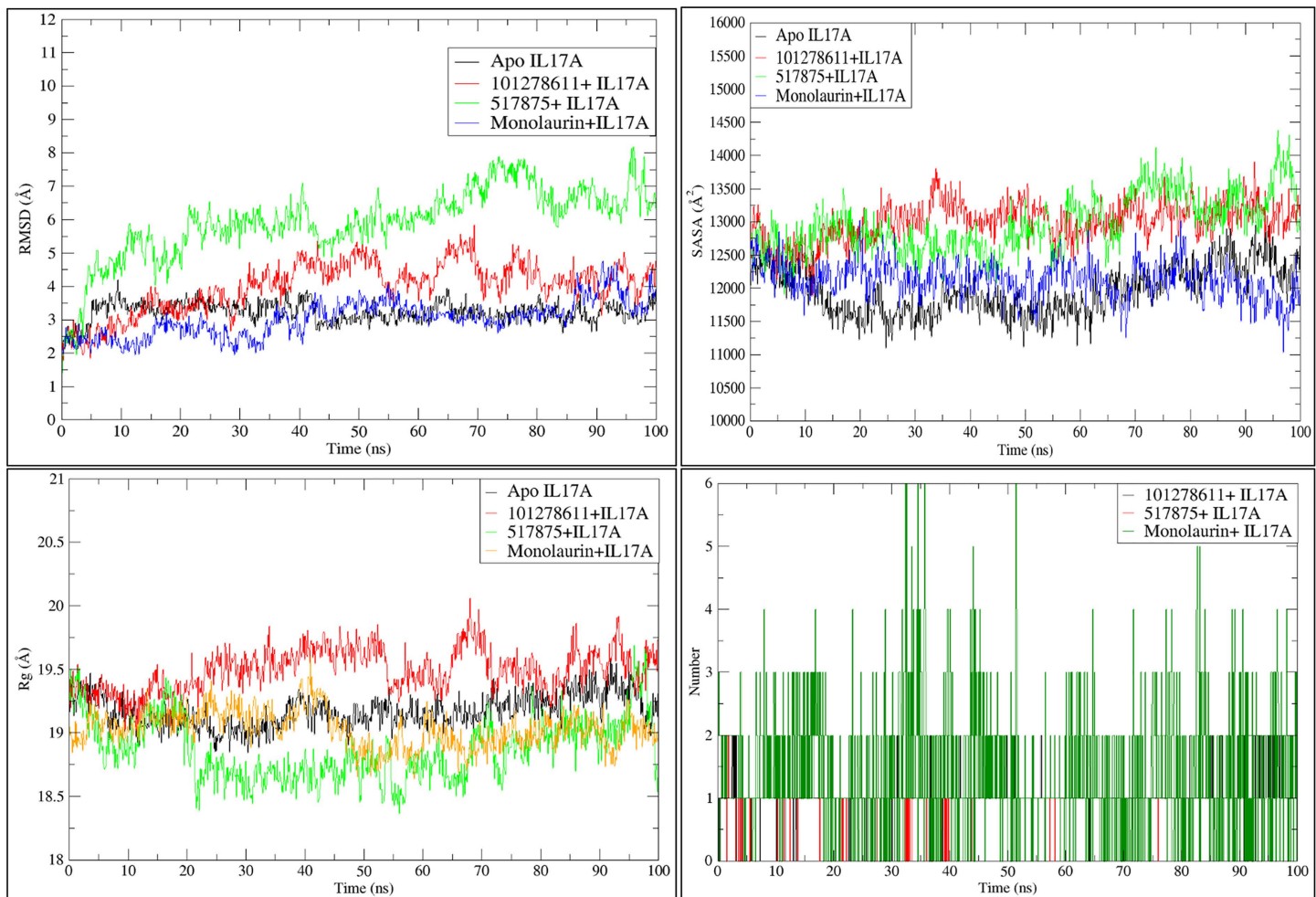

**Fig 2. Representations of simulated trajectory analyses of apo form of IL-17A and ligands complexed with IL-17A.** a) Root mean square deviations (RMSDs) b) Solvent accessible surface area (SASA) c) Radius of gyration (Rg) d) overall H bond occupancy.

residue Leu 97 for an extended period (Fig 2d). The root mean square fluctuation (RMSF) analysis indicated lower fluctuations for most amino acid residues, with compound 517875 complexed with IL-17A showing higher fluctuations in specific regions (Fig 3). Overall, IL-17A complexes exhibited stable binding by forming multiple interactions during the simulation. Hydrogen bond occupancy was examined during the simulation of phytochemical complexes with IL-17A, with the majority of interactions occurring with Leu 97 by all three shortlisted phytochemicals. Specifically, compound 101278611 predominantly interacted with Leu 97 throughout approximately 70% of the entire simulation, while compound 517875 interacted with Leu 97 for up to 15.90% of the 100 ns simulation. Monolaurin interacted with Leu 97 for approximately 38% of the 100 ns simulation, along with various hydrogen bonds with other residues on the binding surface (Fig 4).

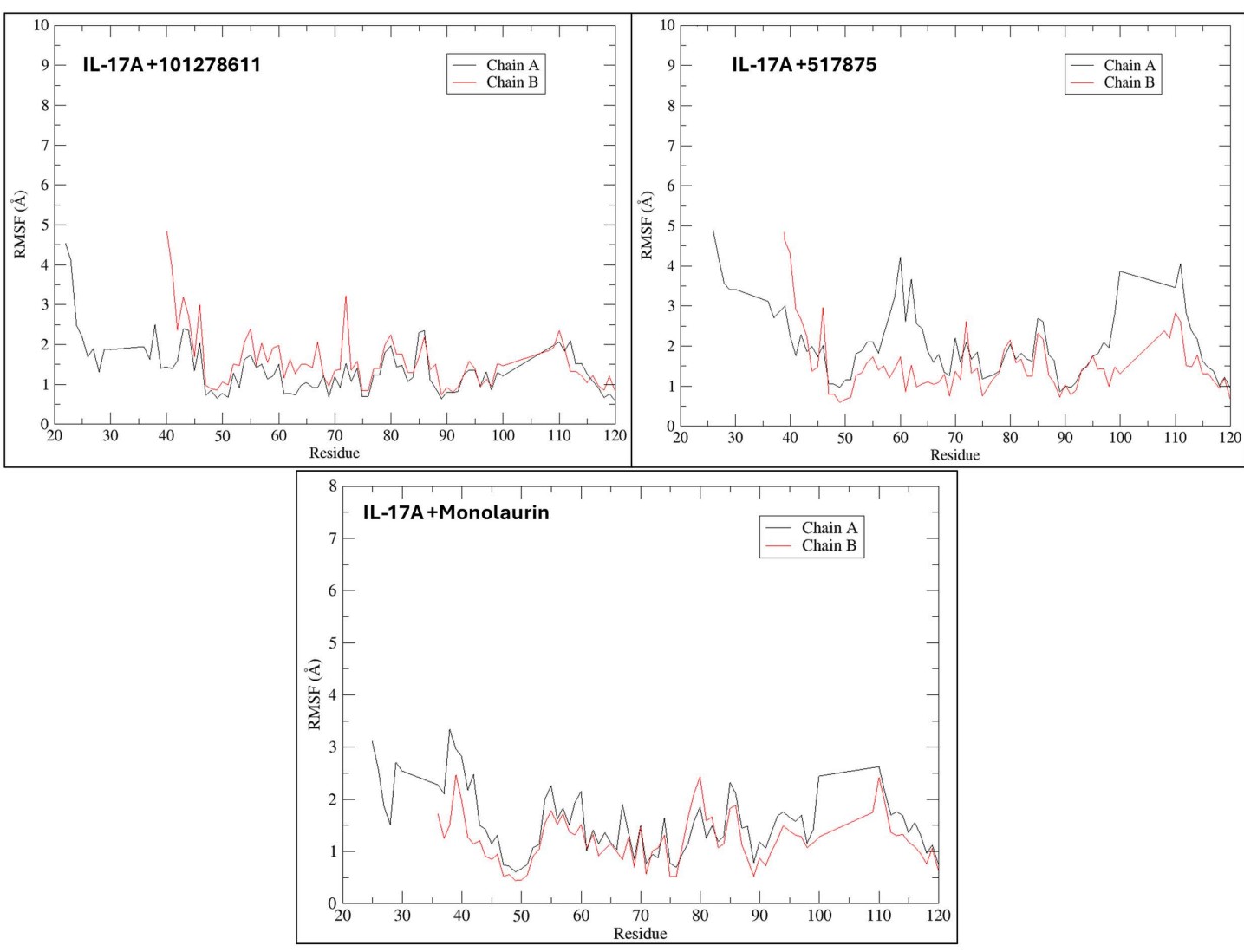

**Fig 3. Representations for root mean square fluctuations (RMSF) of residues in IL17A complexed with respective ligands during simulation for 100 ns.**

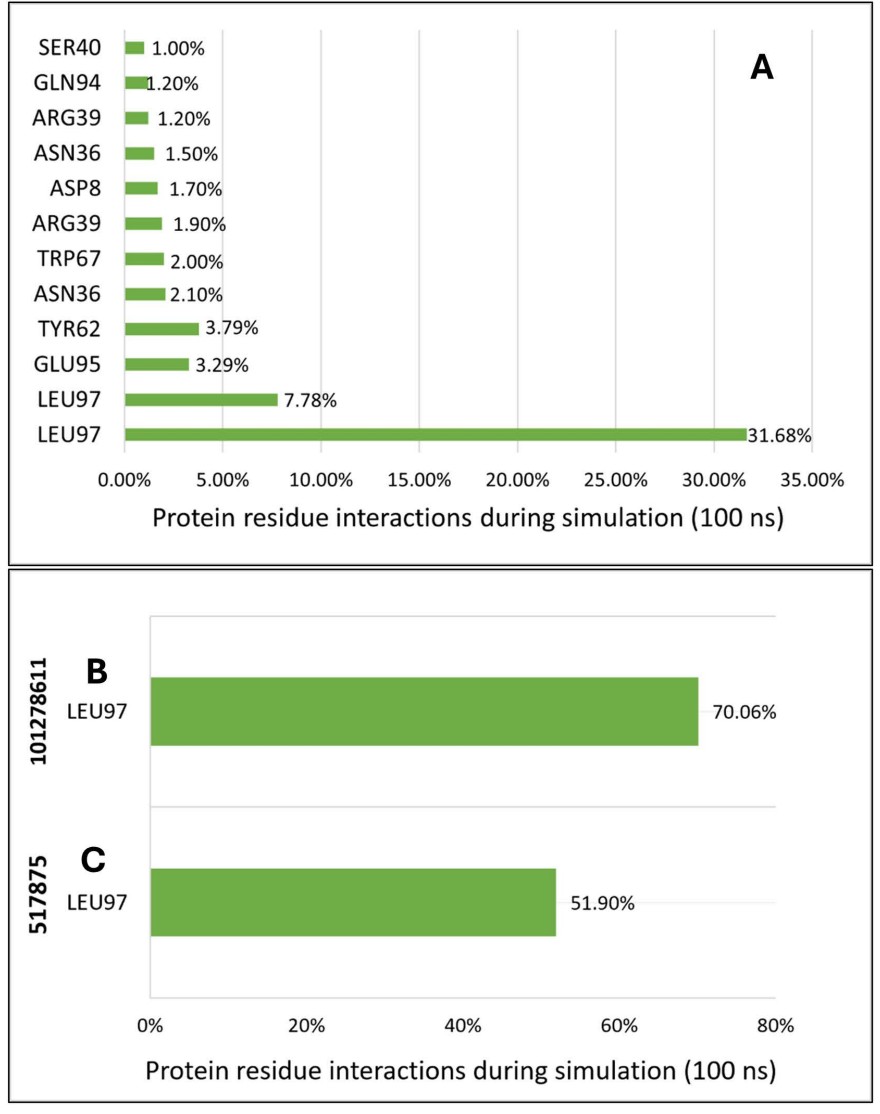

**Fig 4. Representation of residue wise H bond occupancy of all phytochemical + IL17A complexes during 100 ns simulation; a) Monolaurin b) 101278611 c) 517875.**

## Binding energetics profiling of simulated complexes of IL-17A with respective shortlisted compounds using MMPBSA analysis

The components contributing to the binding free energies of individual phytochemical complexes with IL-17A and TNF-α were examined using the Molecular Mechanics Poisson-Boltzmann Surface Area (MM-PBSA) method. Total binding energy, solvation free energy, gas phase energy, non-polar free energy, effective polarizable bonds, electrostatic energy, and van der Waals forces were assessed for each simulated complex. Among the phytochemicals complexed with IL-17A, compound 101278611 exhibited the highest total binding free energy (−38.76 kcal/mol), followed by monolaurin (−23.41 kcal/mol) and compound 517875 (−22.93 kcal/mol). Additionally, electrostatic and van der Waals forces were more prominent for 2,3 101278611 and monolaurin (Fig 5).

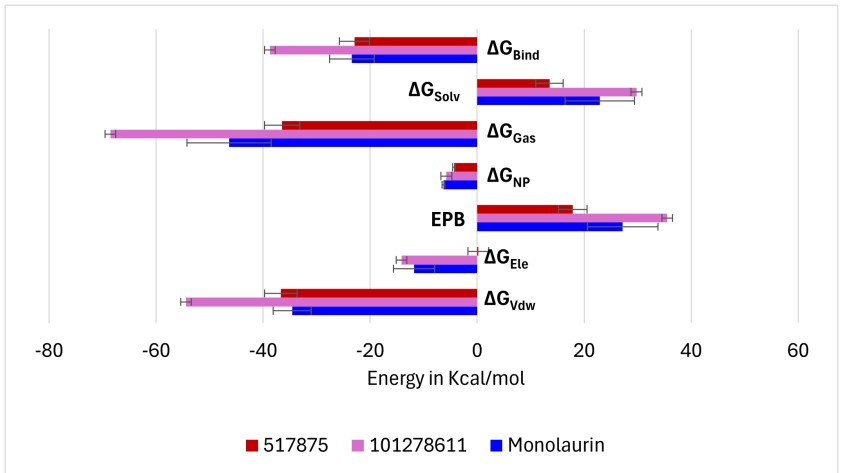

**Fig 5. Components of binding free energies of IL-17A in presence of respective shortlisted phytochemical during MMPBSA analysis.**
$\Delta G_{bind}$ = Total binding free energy; $\Delta G_{solv}$ = Solvation free energy; $\Delta G_{gas}$ = gas-phase energy; $\Delta G_{NP}$ = non-polar solvation free energy; EPB = effective polarizable bonds; $\Delta E_{ele}$ = electrostatic energy; $\Delta G_{vdW}$ = Vander Waal forces.

### Principal Component Analysis (PCA) of simulated complexes of IL-17A with respective shortlisted compounds

Compound 101278611 does not significantly alter IL-17 A's global conformational space but increases its heterogeneity, introducing novel conformations. Compound 517875 induces a major conformational shift, expanding IL-17 A's conformational variability and stabilizing rare states. Monolaurin enhances IL-17 A's flexibility, broadening its conformational distribution and inducing structural shifts upon binding. (Fig 6)

### Eigenvector-based analysis of clusters of confirmations using principal components

For IL-17A, the black curve (apo form) shows the highest flexibility, indicating greater dynamic motion when unbound. Ligand binding reduces fluctuations, with compound 101278611 (red) providing the most stabilization (sharpest initial drop). Compound 517875 (green) and Monolaurin (blue) impose moderate motion restrictions but not as strongly as Compound 101278611. (Fig 7)

### Analysis of the interactions between short-listed ligands of VT oil and the binding site of TNF- α

The native ligand (6,7-Dimethyl-3-[[methyl-[2-[methyl-[[1-[4-(trifluoromethyl) phenyl] indol-3-yl] methyl] amino] ethyl] amino] methyl] chrome-4-one, CID: 101506407) that bound to the TNF-α crystal structure (PDB ID: 2AZ5) engaged in interactions with Gly 121, Tyr 119, Leu 57, Tyr 59, and Tyr 151 residues on the binding surface (S4 Fig, See: S2 File). Similarly, the shortlisted ligands (S7 Fig, See: S2 File) from VT oil docked with TNF-α showcased interactions with the aforementioned residues along with additional residues on the binding surface, including Ser 60, Leu 120, and Gln 61 (Fig 8). Dimyristin, with a minimum binding energy of −9.873 kcal/mol, formed a total of 13 interactions with the TNF-α binding surface. There were three hydrogen contacts with Gln 149, and one each with Tyr 151 and Gln 61 and hydrophobic interactions involved Leu 57, Tyr 59, His 15, Tyr 119, and Tyr 151. Lupeol recorded the second highest number of interactions (12) with a minimum binding energy of −9.409 kcal/mol. This compound formed two hydrogen bonds with Ser 60 and Leu 120, followed by hydrophobic interactions with Tyr 59, Leu 57, Tyr 119, and Tyr 151. Ginsenol demonstrated a total of 7 interactions at a minimum binding energy of −8.694 kcal/mol, including two hydrogen interactions with Gly 121 and Leu 120, and hydrophobic contacts with Tyr 119, Tyr 151, and Tyr 59. Monopalmitin exhibited the least number of interactions (5) with

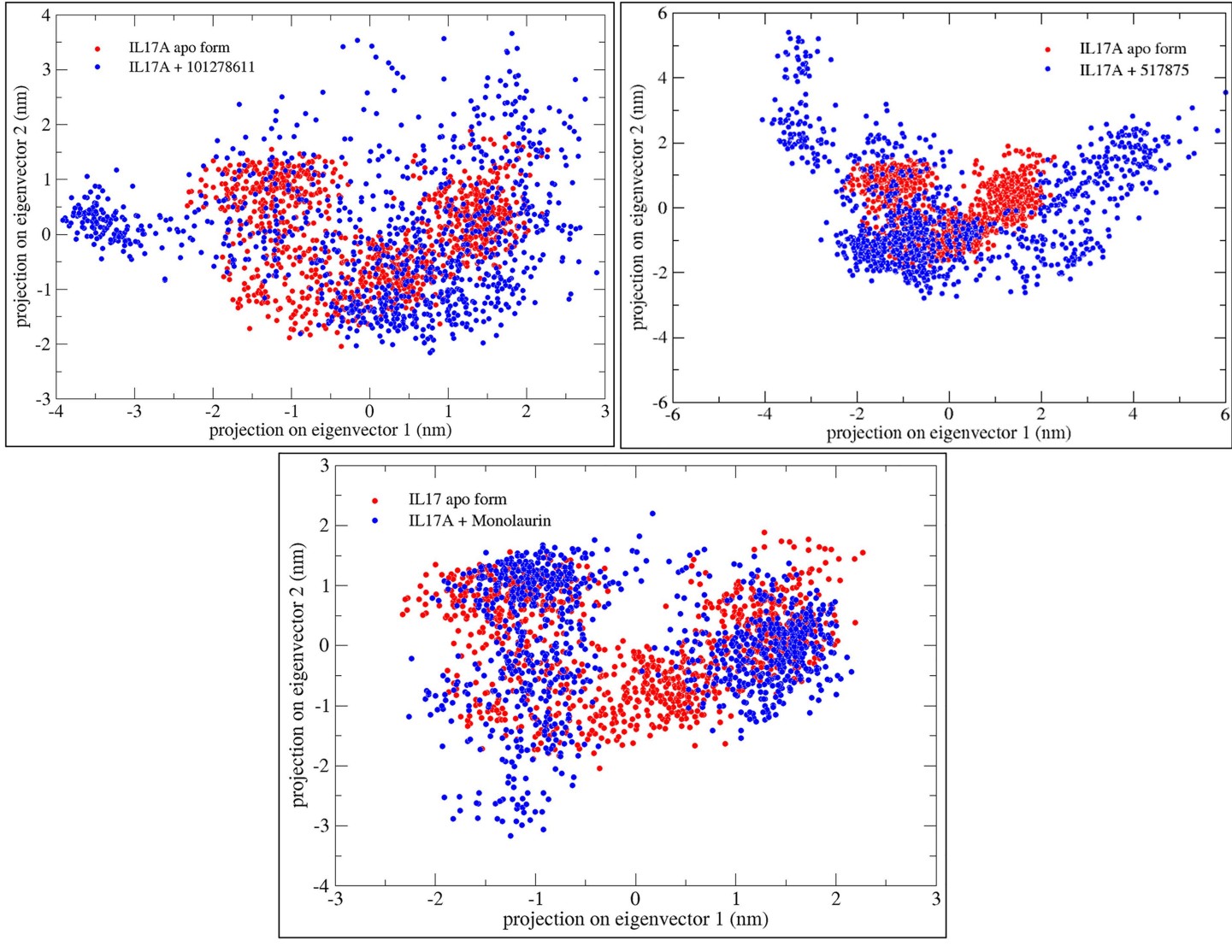

**Fig 6. Representation of 2D projections and analysis of principal components of the trajectories of IL-17A apo form (confirmations represented in red color) and IL-17A in presence of respective ligand (confirmations represented in blue color) from molecular dynamics simulation of 100 ns.**

a minimum binding energy of −8.644 kcal/mol, comprising two hydrogen interactions with Tyr 151 and Ser 60, and hydrophobic interactions with Tyr 119 and Tyr 59 (Table 2).

## Dynamic behavior of TNF-α complexed with respective shortlisted ligands of VT oil during simulation

Post-molecular dynamics (MD), individual complexes of the four chosen phytochemicals (dimyristin, lupeol, ginsenol, and monopalmitin) with TNF-α displayed initial slight upward trends in root mean square deviation (RMSD), reaching stabilization from 50 ns onwards until the completion of the simulation (Fig 9a). The solvent-accessible surface area (SASA) graph for the TNF-α complexes remained consistently lower compared to the apo form (Fig 9b). The radius of gyration (Rg) profiles of the complexes exhibited steady behavior without notable deviations (Fig 9c). Hydrogen bond analysis revealed that dimyristin formed the most

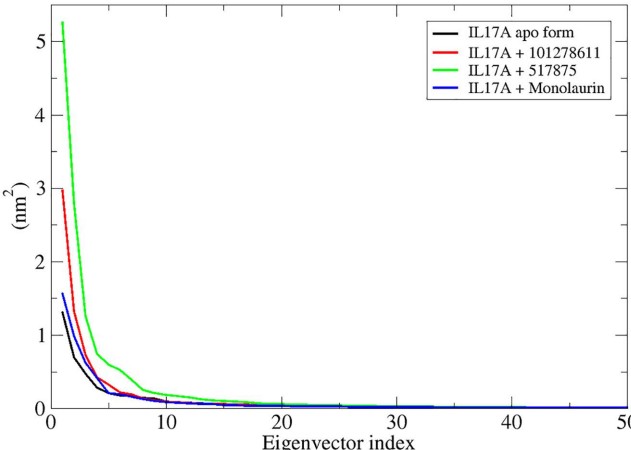

**Fig 7. Representation of eigenvector index of IL17A apo form (black) and IL17A in presence of respective ligand (red, green, blue) from their first 50 eigenvectors versus eigenvalues.**

**Table 2. Shortlisted phytoconstituents of VT and their interactions with TNF – α.**

| Sl. No | Phytoconstituents with structure | Binding energy in kcal/mol | No of interactions | Hydrogen bonds | Hydrophobic bonds |
|---|---|---|---|---|---|
| 1 | Dimyristin ($C_{31}H_{60}O_5$) | −9.873 | 13 | Gln 149 (3)<br>Gln 61<br>Tyr 151 | Leu 57(2)<br>Tyr 59 (2)<br>His 15<br>Tyr 119<br>Tyr 151(2) |
| 2 | Monopalmitin ($C_{19}H_{38}O_4$) | −9.409 | 5 | Tyr 151<br>Ser60 | Tyr 119<br>Tyr 59 (2) |
| 3 | Lupeol ($C_{30}H_{50}O$) | −8.694 | 12 | Ser 60<br>Leu 120 | Tyr 59 (3)<br>Leu 57(2)<br>Ile 155 (2)<br>Tyr 119 (2)<br>Tyr 151 |
| 4 | Ginsenol ($C_{15}H_{26}O$) | −8.644 | 7 | Leu 120<br>Gly 121 | Tyr 119 (3)<br>Tyr 151<br>Tyr 59 |

hydrogen bonds, followed by ginsenol, lupeol, and monopalmitin (Fig 9d). Root mean square fluctuations (RMSF) of the complexes indicated generally lower fluctuations among most amino acid residues, with higher fluctuations observed for Gln88 (Fig 10). Like the IL-17A complexes, TNF-α complexes maintained binding rigidity during the simulation. Hydrogen bond occupancy during the simulation predominantly involved key residues including Gly 121, Tyr 119, Leu 57, Tyr 59, and Tyr 151, as well as Ser 60, Leu 120, and Gln 61 of the binding surface. Dimyristin interacted with Tyr 151 for approximately 33.89% of the simulation period. Similarly, lupeol maintained interaction with Tyr 119 throughout the simulation, followed by ginsenol, which formed interactions with Tyr 59 for up to 25.35%, and monopalmitin with 15.37% during the 100 ns simulation period (Fig 11).

## Binding energetics profiling of simulated complexes of TNF-α with respective shortlisted compounds using MMPBSA analysis

Dimyristin, among the shortlisted phytochemicals complexed with TNF-α, achieved the highest total binding free energy of −45.03 kcal/mol, followed by lupeol at −31.21 kcal/mol, monopalmitin at −28.47 kcal/mol, and ginsenol at −18.11 kcal/mol.

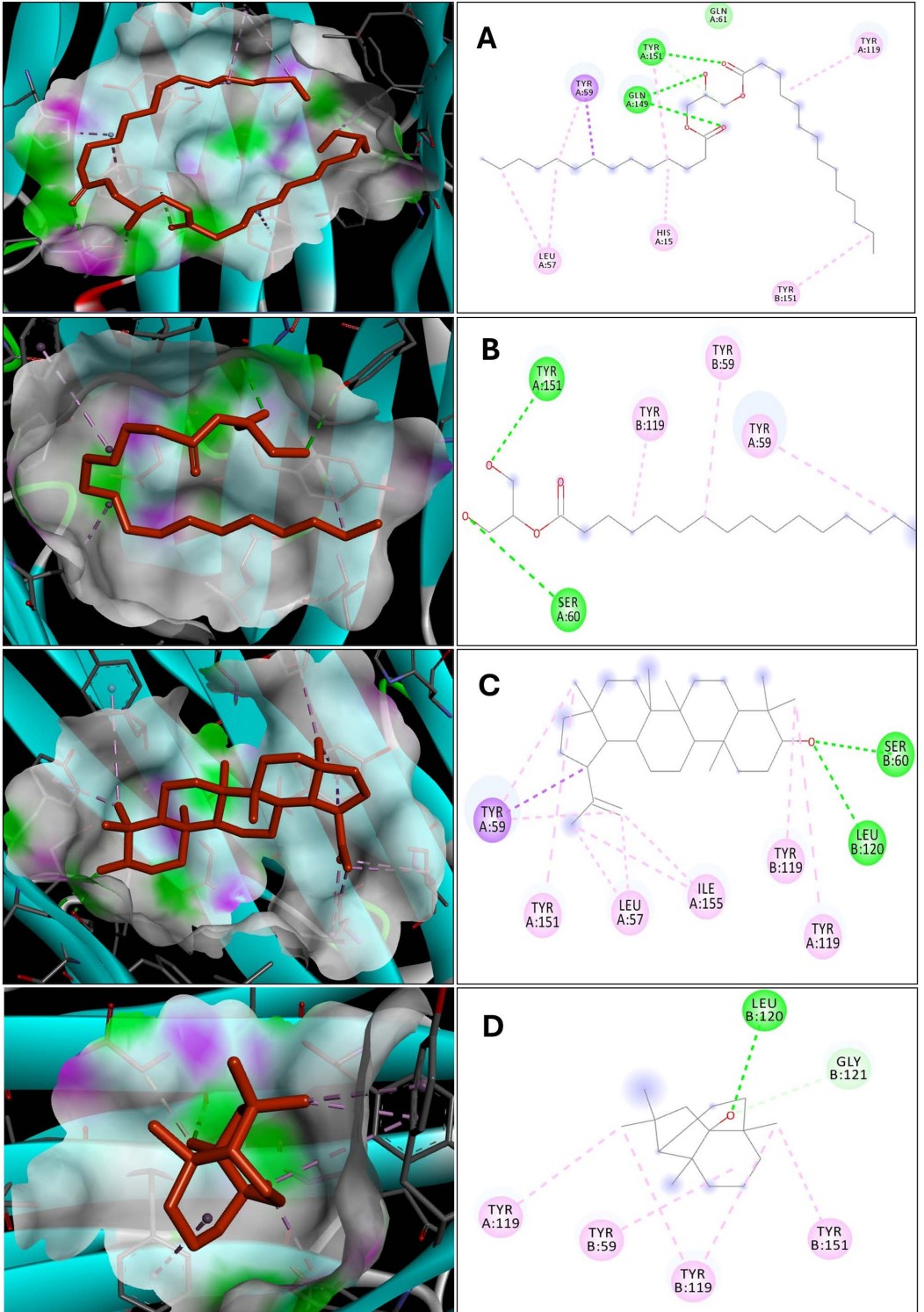

**Fig 8. Display of docking interactions made by the shortlisted phytoconstituents of VT with the binding site of TNF-α.** Light and dark green lines: hydrogen interactions (<3 Å distance); purple & pink lines: hydrophobic interactions. A) Dimyristin B) Monopalmitin C) Lupeol D) Ginsenol.

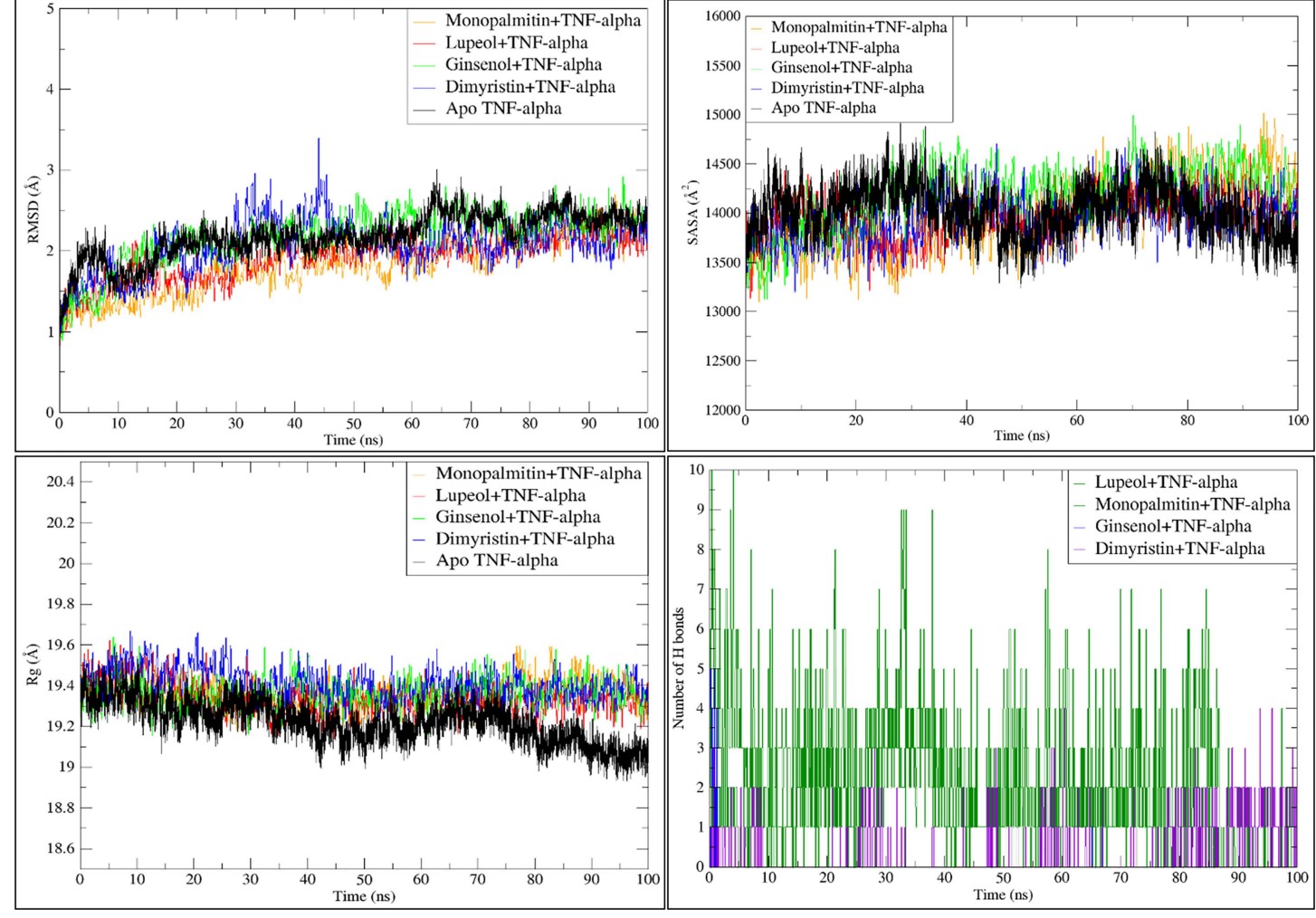

**Fig 9. Representations of simulated trajectory analyses of apo form of TNF-α and ligands complexed with TNF-α.** a) Root mean square deviations (RMSDs) b) Solvent accessible surface area (SASA) c) Radius of gyration (Rg) d) Overall H bond occupancy.

Electrostatic forces were observed to contribute less compared to van der Waals forces in all simulated TNF-α complexes. Similarly, in line with the total binding free energy, the dimyristin-TNF-α complex exhibited the highest van der Waals forces, followed by lupeol and monopalmitin, surpassing ginsenol. (Fig 12)

### Principal Component Analysis (PCA) of simulated complexes of TNF-α with respective shortlisted compounds

Monopalmitin binding largely preserves TNF-α's conformational space but may induce localized structural changes or stabilization. Ginsenol shows significant overlap with the apo form, indicating minimal impact on TNF-α dynamics. Lupeol also overlaps considerably but forms distinct clusters, suggesting stabilization of specific conformations and reduced flexibility. Dimyristin exhibits similar behavior, with distinct clusters indicating stabilization of less-populated states, potentially restricting flexibility and influencing biological activity. (Fig 13)

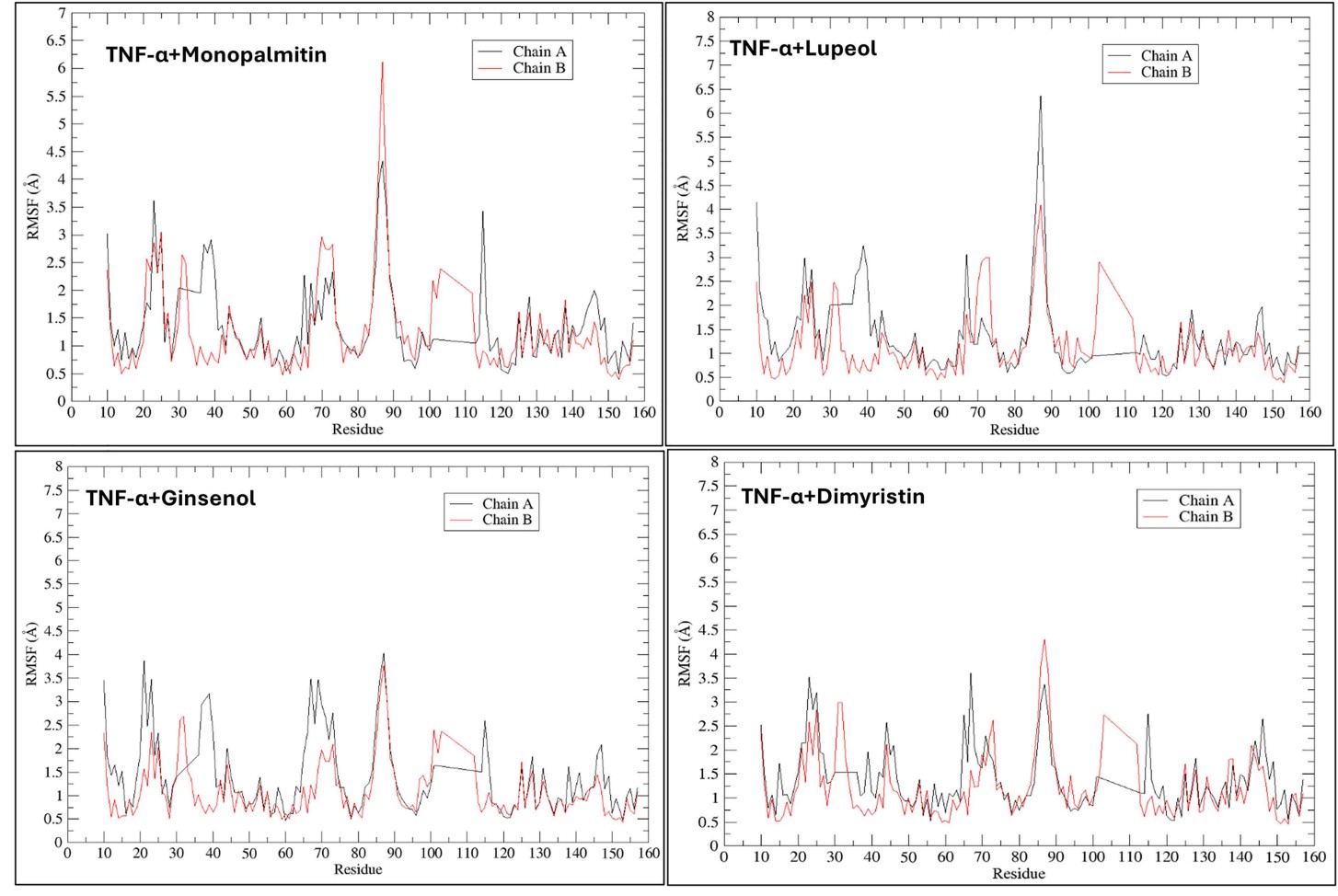

**Fig 10. Representations of root mean square fluctuations (RMSF) of residues in TNF-α complexed with respective ligands during simulation for 100 ns.**

### Eigenvector-based analysis of clusters of confirmations of TNF-α with respective shortlisted compounds

Eigenvector analysis reveals binding-induced conformational constraints, shedding light on ligand effects on TNF-α stability and function. TNF-α bound to Dimyristin (red) and Ginsenol (blue) shows slightly reduced fluctuations in dominant eigenvectors compared to the apo form, suggesting stabilization. Lupeol (magenta) and Monopalmitin (green) exhibit minor deviations, indicating similar stabilizing effects with slight flexibility differences. (Fig 14)

### Physical characterization, drug-likeliness and ADMET analyses of shortlisted phytochemicals from VT oil

Most compounds, including monolaurin, lupeol, and monopalmitin, adhere to Lipinski's Rule, suggesting good oral bioavailability. Dimyristin exceeds molecular weight and hydrogen bond limits, making it less suitable for oral use (S2 Table, See: S3 File). Lupeol was predicted to function as a nuclear receptor ligand and enzyme inhibitor, while 517875 and 101278611 predicted to have protease and enzyme inhibition properties. Other compounds exhibit minimal bioactivity (S3 Table, See: S3 File). Monolaurin, compound 517875, monopalmitin, and ginsenol predicted to exhibit high GI absorption, supporting oral bioavailability. Monolaurin and lupeol predicted to cross the BBB (S4 Table, See: S3 File). Lupeol was predicted to have high clearance rate, while Monolaurin shows moderate clearance rate, indicating efficient elimination.

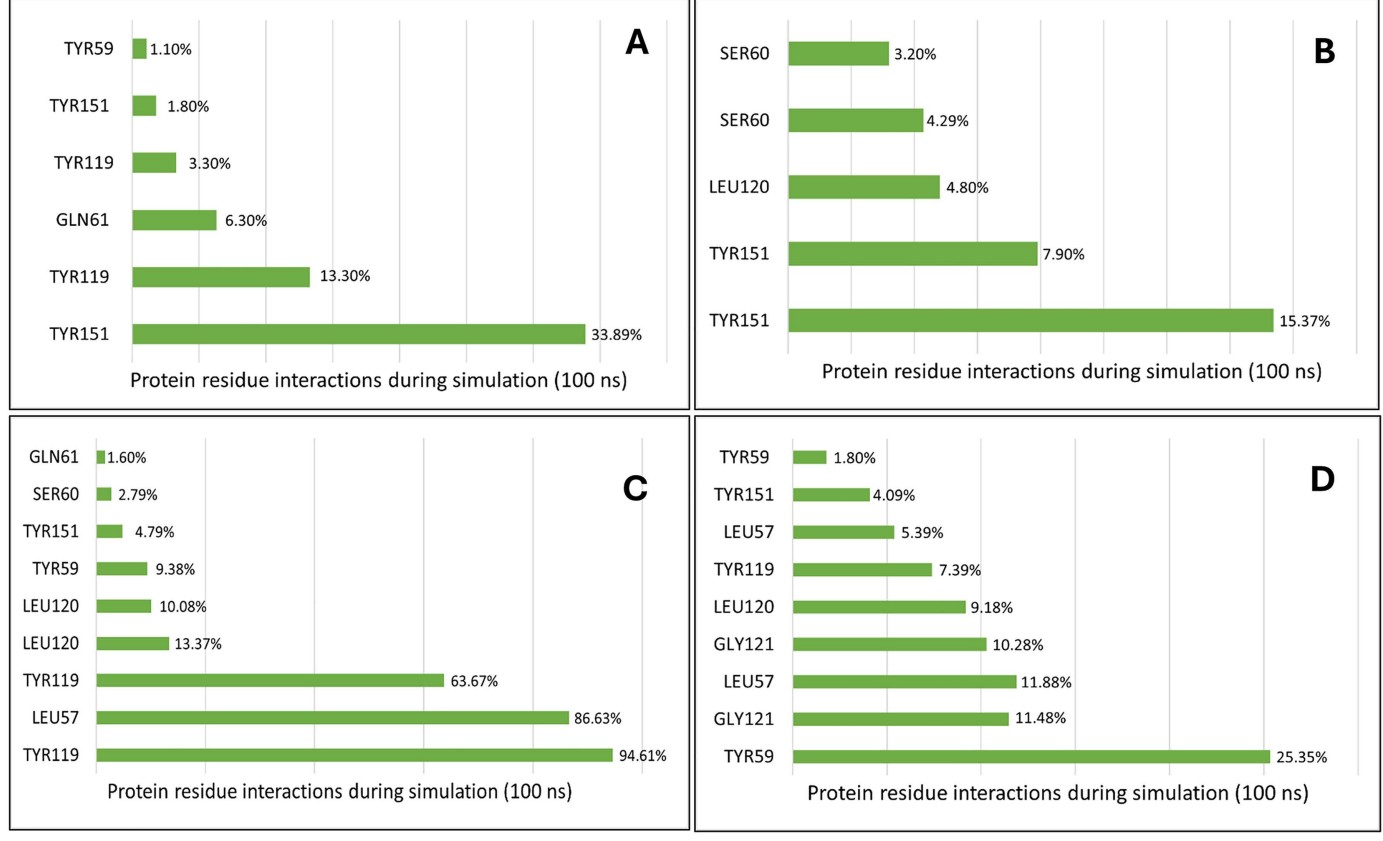

**Fig 11. Representation of residue wise H bond occupancy of all phytochemicals +TNF-α complexes during 100 ns simulation; a) Dimyristin b) Monopalmitin c) Lupeol d) Ginsenol.**

Dimyristin is predicted to be carcinogenic, whereas other compounds exhibit lower toxicity profiles, confirmed by negative AMES tests and high LD50 values (>2000 mg/kg) (S5 Table, See: S3 File). Most of the analyzed compounds exhibit low drug-likeness scores, implying potential challenges in their pharmacokinetic properties (S8 Fig, See: S2 File).

## Discussion

This research explores the anti-inflammatory properties of *Vetpalai thailam*, a Siddha medicinal oil used for skin conditions, particularly psoriasis. It examines how its bioactive compounds influence proinflammatory cytokines TNF-α and IL-17A. The study employs GC-MS phytochemical analysis and pharmacoinformatics techniques to screen bioactive substances based on molecular interactions, pharmacokinetics, and drug likeness scores [27]. Several promising compounds were identified and validated through in-silico experiments, offering insights into potential therapeutic interventions [28]. Based on the research findings from the literature, it was hypothesized that this oil contains numerous anti-inflammatory compounds effective in alleviating psoriatic inflammation [29]. The Inhibition of IL-17A and TNF-α was observed to improve the condition of psoriasis and enhance quality of life [30]. The inflammatory pathway was further investigated, and proinflammatory cytokines TNF-α and Il-17A were selected for a pharmacoinformatics study using all 65 compounds of VT oil.

Based on the premise, an effort was undertaken to elucidate the impact of identified phytochemicals on the aforementioned cytokines using molecular modeling techniques. Molecular docking and dynamics analyses revealed the binding potential of phytochemicals in Vetpalai thailam (VT) oil. Among the selected compounds, dimyristin emerged as the primary candidate, chemically classified as a diacylglycerol containing myristic acid. Notably, a study highlights the

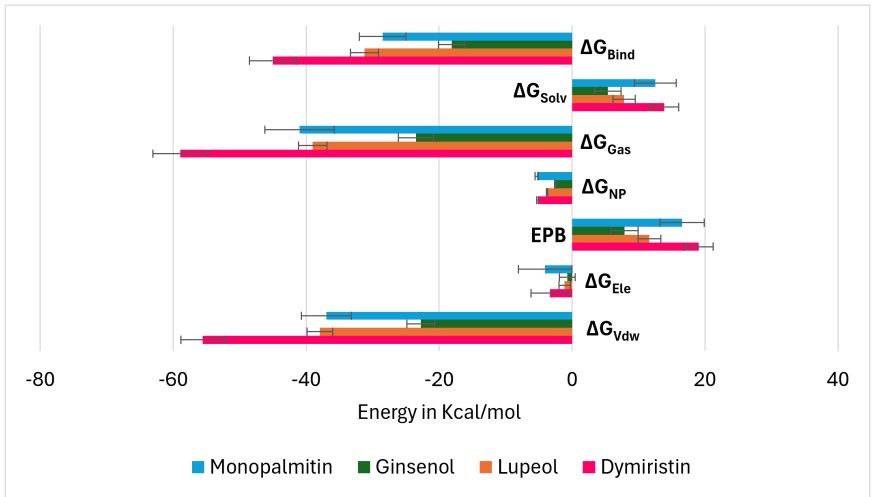

**Fig 12. Components of binding free energies of TNF-α complexed with respective shortlisted phytochemical during MMPBSA analysis.**
$\Delta G_{bind}$ = Total binding free energy; $\Delta G_{solv}$ = Solvation free energy; $\Delta G_{gas}$ = gas-phase energy; $\Delta G_{NP}$ = non-polar solvation free energy; EPB = effective polarizable bonds; $\Delta E_{ele}$ = electrostatic energy; $\Delta E_{vdW}$ = Vander Waals force.

presence of acylated lysine residues immediately downstream of a hydrophobic, likely membrane-spanning segment of the propiece. The specific myristyl acylation of the TNF-α propiece may enhance its membrane insertion or anchoring, playing a crucial role in inflammatory mediation [31]. Lupeol, a pentacyclic triterpenoid, was identified as another key phytochemical, known for its anti-inflammatory and skin tissue repair properties. Studies indicate that lupeol achieves these effects through the modulation of TNF-α, NF-κB, and Ki-67 [32] (S6 Table, See: S3 File). Similarly, ginsenol, a tricyclic sesquiterpene alcohol found in ginseng rootlets, exhibits immunomodulatory properties by influencing TLR activation. Various ginseng extracts have been shown to reduce TNF-α, IL-1β, IL-6, IFN-γ, IL-12, and IL-18 levels in Staphylococcus aureus-challenged mice, significantly mitigating inflammatory responses compared to untreated controls [33]. Another study further supports the ability of ginseng extract to suppress pro-inflammatory cytokine production, thereby alleviating symptoms and progression of inflammatory diseases [34]. Monopalmitin, a monoglyceride, was the final phytochemical identified with binding capability against TNF-α. While its direct anti-inflammatory properties remain unexplored, plant extracts rich in monopalmitin have demonstrated notable anti-inflammatory effects [35].

On the other hand, three phytochemicals exhibited minimal binding energies and distinct interactions with IL-17A (Table 1). Among them, monolaurin demonstrated the strongest affinity, showing the highest number of interactions and the lowest binding energy in both docking and simulation analyses. As a monoglyceride, monolaurin has been reported to mitigate *E. coli*-induced intestinal inflammation in piglets by inhibiting the NF-κB/MAPK pathways, highlighting its potential anti-inflammatory effects [36]. Next, trimethylsilyl 2,4-bis(trimethylsilyloxy)benzoate (517875) emerged as a notable phytochemical, recognized for its antibacterial and antifungal properties, as acknowledged by multiple studies. However, its full therapeutic potential remains largely unexplored [37]. The 2,3-bis(trimethylsilyloxy)propyl (9E,12E)-octadeca-9,12-dienoate (101278611) was identified in the VT oil sample, representing a novel addition to the chemical database. Due to scarce information available in the PubChem compound database, further investigation is required to understand its pharmacological significance.

Molecular docking, simulation dynamics, PCA and eigenvector analyses were conducted to assess the binding affinities of selected VT oil phytochemicals against IL-17A and TNF-α. A pivotal cytokine IL-17A has gained traction as a therapeutic strategy in pathogenesis of autoimmune and inflammatory conditions such as psoriasis. Structural and computational

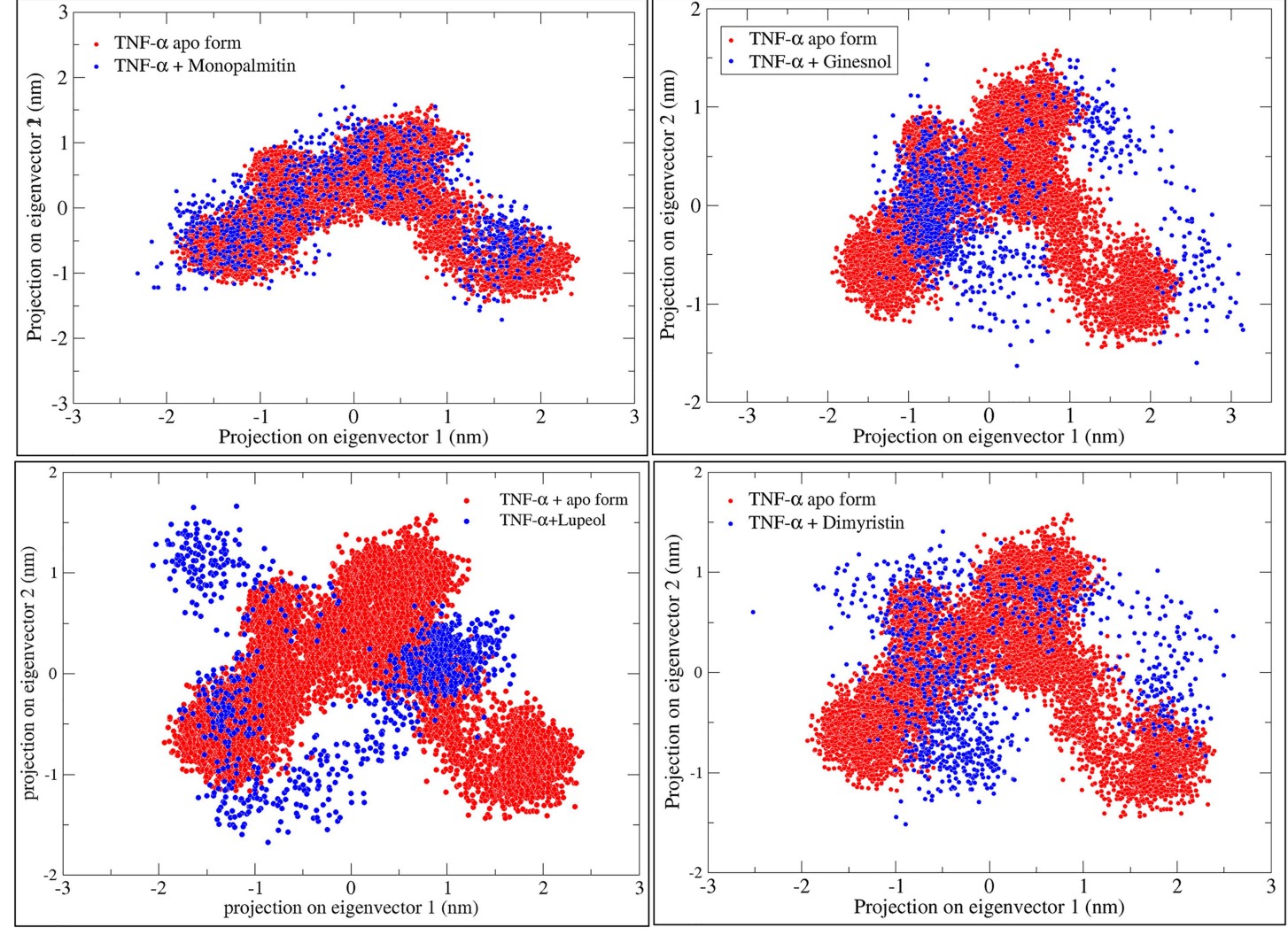

**Fig 13. Representation of 2D projections and analysis of principal components of the trajectories of TNF-α apo form (confirmations represented in red colour) and TNF-α in presence of respective ligand (confirmations represented in blue colour) from molecular dynamics simulation of 100 ns.**

analyses have identified a conserved hydrophobic pocket at the IL-17A dimer interface, comprising residues such as Pro63, Tyr62, Ile96, Leu97, Val98, Leu99, and Leu112, (S5 Fig, See: S2 File) which facilitate critical van der Waals and π–π stacking interactions with both macrocyclic inhibitors and [38,39]. Biologics such as netakimab and secukinumab also exploit this epitope cluster, particularly Leu97 and Pro63 [40]. Leu97 and Leu112 are particularly significant, contributing to ligand anchoring and structural adaptability. These residues overlap with IL-17RA binding sites, underscoring their pharmacological relevance [38–40].

In this study, VT oil derived phytochemicals notably monolaurin and compound 101278611 demonstrated strong binding to IL-17A. Compounds 101278611 and 517875 also showed strong binding affinities, engaging Leu97 and Pro63 through varying interaction patterns (Table 1). Monolaurin formed multiple hydrogen bonds with Leu97, while compound 101278611 exhibited the most favorable binding free energy (−38.76 kcal/mol). Principal component and eigenvector

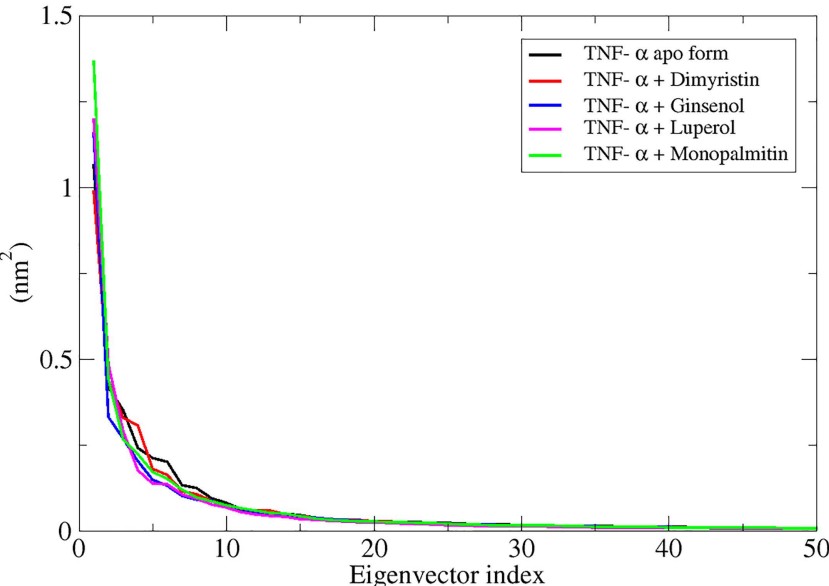

**Fig 14. Representation of eigenvector index of TNF-α apo form (black) and TNF-α in presence of respective ligand (red, blue, magenta, green) from their first 50 eigenvectors versus eigenvalues.**

analyses confirmed stable, biologic-like conformational effects, supporting the potential of these natural compounds as structurally validated IL-17A inhibitors (Figs 2-7)

Tumor necrosis factor-alpha (TNF-α) inhibition relies on a conserved cluster of residues Leu57, Tyr59, Ser60, Gln61, Tyr119, Leu120, Gly121, and Tyr151 located at the homotrimer interface, forming a hydrophobic and polar landscape essential for therapeutic targeting [40]. In addition, we observed the reference (native) ligand interacted with Gly121, Tyr119, Leu57, Tyr59, and Tyr151 (S3 and S4 Figs, See: S2 File). These residues mediate π-alkyl, hydrogen bonding, and hydrophobic interactions with small molecules (e.g., SPD304), peptides, and biologics, contributing to trimer destabilization and receptor blockade [41]. Residues Tyr119 and Gly121 are particularly critical for ligand anchoring and interface stability [41]. Biologics such as infliximab and adalimumab bind overlapping interface loops, including Tyr119 and Leu120, effectively blocking TNFR1/2 interactions [42]. VT oil derived phytochemicals mimic these mechanisms. Dimyristin showed the strongest binding affinity, forming 13 interactions and achieving a docking energy of –9.873 kcal/mol, followed by lupeol and ginsenol (Table 2). Molecular dynamics and MM-PBSA analyses confirmed complex stability and favorable energetics (ΔG_bind = –45.03 kcal/mol) (Fig 12). Principal component and eigenvector analyses (Figs 13 and 14). revealed ligand-induced stabilization, with dimyristin and lupeol reducing TNF-α flexibility. These findings suggest that VT phytochemicals, particularly dimyristin, structurally and energetically resemble other experimentally validated inhibitors [43,44] supporting their potential to be natural anti-inflammatory agents in conditions like psoriasis.

This study critically examines seven bioactive compounds by analyzing their physicochemical and ADMET properties to determine pharmacological relevance. Parameters such as molecular weight, TPSA, rotatable bonds, hydrogen bonding, GI absorption, BBB permeability, and toxicity were assessed. Ginsenol and lupeol, with low TPSA and minimal rotatable bonds, show strong potential for CNS therapies due to their ability to cross the BBB and may influence CNS functions. Lupeol also acts as a nuclear receptor ligand and enzyme inhibitor, though its high clearance and short half-life may require frequent dosing. Monolaurin and 2-Monopalmitin demonstrate high GI absorption and moderate clearance, making them suitable for oral delivery. Compound 517875 shows protease inhibition and favorable absorption. In contrast, Dimyristin and 101278611 have high molecular weights and excessive rotatable bonds, limiting oral bioavailability and

suggesting topical use. This may be a key reason for the external use of VT oil in Siddha Medicine. Dimyristin's carcinogenicity further restricts its pharmaceutical application. Despite varied bioactivity, most compounds exhibit low toxicity, supported by negative AMES tests and $LD_{50}$ values above 2,000 mg/kg. These findings highlight diverse pharmacological potential and the need for tailored formulation strategies and structural optimization to enhance delivery, efficacy and safety (S4 and S5 Tables, See, S3 File).

## Conclusion

This study demonstrates the potential of phytochemicals derived from VT oil as modulators of IL-17A and TNF-α, two central mediators of inflammatory responses. Pharmacoinformatics analyses revealed strong binding affinities and stabilizing interactions, with special emphasis of monolaurin, compound 101278611, and dimyristin emerging as the most promising candidates. These compounds exhibited interaction profiles comparable to those of established biologics, as evidenced by crystallographic structural data of the target cytokines. Additional support from native ligand interactions and findings from related studies that showcased the inhibitory abilities of small molecules further substantiates cytokine-inhibiting capabilities of shortlisted ligands of VT oil. To boot, free energy calculations and principal component analysis (PCA) reinforced the stability and functional relevance of these interactions. Despite their positives, some compounds displayed low drug-likeness scores, indicating potential pharmacokinetic limitations. To address these challenges, QSAR modeling and lead optimization strategies are recommended to enhance their drug delivery and pharmacokinetic profiles. Overall, these findings provide a compelling basis for further in-vitro and in-vivo investigations to fully elucidate the therapeutic potential of VT oil-derived compounds in managing inflammatory disorders, including psoriasis.

## Supporting information

**S1 File. GC-MS report detailing the identification and characterization of phytochemical constituents present in the VT oil sample.**
(PDF)

**S2 File. Figures S1–S8 supporting molecular docking validation, interactions of native ligands with respective proteins, 2D structural representations of shortlisted compounds, and their drug-likeness profiles.**
(PDF)

**S3 File. Tables S1–S6 summarizing phytochemical data from GC-MS analysis, ADMET and bioactivity predictions of shortlisted compounds, and literature-based pharmacological insights into some of the identified phytoconstituents.**
(PDF)

## Acknowledgments

All authors are grateful to Sivasakthi Pharmaceutical Pvt Ltd, Coimbatore, Tamil Nadu, India for providing us with the *Vetpalai thailam* (Siddha formulation) to carry out the research and to Analytical Research & Metallurgical Laboratories Pvt. Ltd. (ARML), Bengaluru, Karnataka, India for providing the facility to perform the GCMS analysis.

## Author contributions

**Conceptualization:** Nayak Deeksha Dayanand, Shama Prasada Kabbekodu, Arul Amuthan, Vasudha Devi, Rajasekhar Chinta.

**Data curation:** Nayak Deeksha Dayanand.

**Formal analysis:** Nayak Deeksha Dayanand.

**Investigation:** Nayak Deeksha Dayanand, Sathish Pai B, Vasudha Devi, Rajasekhar Chinta.

**Methodology:** Shama Prasada Kabbekodu, Arul Amuthan, Vasudha Devi, Rajasekhar Chinta.

**Project administration:** Rajasekhar Chinta.

**Resources:** Nayak Deeksha Dayanand, Arul Amuthan, Vasudha Devi.

**Software:** Rajasekhar Chinta.

**Supervision:** Sathish Pai B, Vasudha Devi, Rajasekhar Chinta.

**Validation:** Vasudha Devi, Rajasekhar Chinta.

**Visualization:** Rajasekhar Chinta.

**Writing – original draft:** Nayak Deeksha Dayanand, Rajasekhar Chinta.

**Writing – review & editing:** Shama Prasada Kabbekodu, Arul Amuthan, Sathish Pai B, Vasudha Devi, Rajasekhar Chinta.

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
