## [Decision Letter · Decision Letter 0]

14 May 2025

Dear Dr. Chinta,

Thank you for submitting your manuscript to PLOS ONE. After careful consideration, we feel that it has merit but does not fully meet PLOS ONE’s publication criteria as it currently stands. Therefore, we invite you to submit a revised version of the manuscript that addresses the points raised during the review process.

**The submission deserves another round of substantial revision to address some critical concerns raised by a reviewer.. **

We look forward to receiving your revised manuscript.

Kind regards,

Yusuf Oloruntoyin Ayipo, Ph.D

Academic Editor

PLOS ONE

**Journal Requirements:**

1. When submitting your revision, we need you to address these additional requirements. Please ensure that your manuscript meets PLOS ONE's style requirements, including those for file naming. The PLOS ONE style templates can be found at https://journals.plos.org/plosone/s/file?id=wjVg/PLOSOne_formatting_sample_main_body.pdf and https://journals.plos.org/plosone/s/file?id=ba62/PLOSOne_formatting_sample_title_authors_affiliations.pdf 2. Please note that PLOS ONE has specific guidelines on code sharing for submissions in which author-generated code underpins the findings in the manuscript. In these cases, we expect all author-generated code to be made available without restrictions upon publication of the work. Please review our guidelines at https://journals.plos.org/plosone/s/materials-and-software-sharing#loc-sharing-code and ensure that your code is shared in a way that follows best practice and facilitates reproducibility and reuse. 3. In the online submission form, you indicated that “Any additional data can be provided on request, as the other components of research related to this drug is still on going.” All PLOS journals now require all data underlying the findings described in their manuscript to be freely available to other researchers, either a. In a public repository, b. Within the manuscript itself, or c. Uploaded as supplementary information.This policy applies to all data except where public deposition would breach compliance with the protocol approved by your research ethics board. If your data cannot be made publicly available for ethical or legal reasons (e.g., public availability would compromise patient privacy), please explain your reasons on resubmission and your exemption request will be escalated for approval. 4. We notice that your supplementary figures are uploaded with the file type 'Figure'. Please amend the file type to 'Supporting Information'. Please ensure that each Supporting Information file has a legend listed in the manuscript after the references list.

**Additional Editor Comments:**

The submission deserves another round of substantial revision to address some critical concerns raised by a reviewer.

Reviewers' comments:

Reviewer's Responses to Questions

**Comments to the Author**

1. Is the manuscript technically sound, and do the data support the conclusions?

Reviewer #1: Yes

Reviewer #2: Yes

Reviewer #3: Yes

2. Has the statistical analysis been performed appropriately and rigorously?

Reviewer #1: Yes

Reviewer #2: N/A

Reviewer #3: N/A

3. Have the authors made all data underlying the findings in their manuscript fully available?

Reviewer #1: Yes

Reviewer #2: Yes

Reviewer #3: Yes

4. Is the manuscript presented in an intelligible fashion and written in standard English?

Reviewer #1: Yes

Reviewer #2: No

Reviewer #3: Yes

**Reviewer #1:**  The manuscript is well structured with a robust methodology. The combination of both experimental chemical profiling via GC-MS and a well-established in silico analysis, including reliable MD simulations, standard docking procedures, and MMPBSA energy calculations, provides a solid basis for identifying the active ingredients in VT.

The pharmacological relevance of the study is further highlighted by its focus on TNF-α and IL-17A, which are two clinically validated targets in psoriasis. Furthermore, the authors provided insightful discussion on the pharmacological relevance of the key phytochemicals.

A few points to consider to strengthen the robustness of the discussion:

1. Provide more details about the GC-MS compound identification procedure. In the NIST database, what match score threshold was applied? Did any annotations match authentic standards or undergo manual verification?

2. The ADMET and Lipinski evaluations are informative, but there should be more discussion about the relevance of the formulation route. For instance, dimyristin exhibits poor oral bioavailability and fails Lipinski's criteria despite its strong binding affinity. It may be better suited for topical formulations, especially given the traditional mode of VT application.

Overall, the manuscript presents a solid pharmacoinformatics analysis of a conventional antipsoriatic formulation.

**Reviewer #2:**  In this manuscript, Dayanand et al. reports the analysis of isolated chemicals from an oil with anti psoriasis potentials. Authors detailed interaction of select compounds with inflammatory proteins IL-17A and TNF-α. The findings reported are of scientific value that would be of interest to readers, however the manuscript would benefit greatly from the following recommendations.

1. Grammar editing- Sentences in the introduction section were difficult to read.

2. Unit of retention time and mol. wt was not given in Table S1. This should be corrected.

3.Image resolutions are low making it difficult to see important details. Authors should replace images with higher resolution files.

4. Figure legend should follow figures in the result section for better readability.

5. For the analysis of the observed interaction, there was no explanation made on the significance of the observed interactions with the amino acids. Do these interactions have any potential of translating into desirable phenotypic response? If yes, are there basis for such conclusions?

6. Authors should clarify how they arrived at the conclusion on the pharmacological properties and biological activities of selected compounds. Are the stated activities in relation to the studied respective targets?

7. Authors should give detailed citations of literature precedence to support claims on the biological potentials of select compounds, considering that in silico studies are very preliminary.

**Reviewer #3: ** The current study presents a comprehensive phytochemical and pharmacoinformatics analysis of a traditional antipsoriatic oil formulation, with a focus on its potential activity against the proinflammatory cytokines TNF-α and IL-17A. The introduction is well-structured, and the results and discussion are clearly presented. However, the authors should provide a rationale for selecting TNF-α and IL-17A over other cytokines commonly associated with psoriasis. For the molecular docking and molecular dynamics simulations, it is recommended to include appropriate reference or standard controls and compare the performance of the top hits against these benchmarks. Additionally, the quality of the figures is suboptimal, please provide high-resolution versions to ensure clarity and better interpretation.

**Do you want your identity to be public for this peer review?** For information about this choice, including consent withdrawal, please see our Privacy Policy

Reviewer #1: No

Reviewer #2: No

Reviewer #3: No

---

## [Author Response · Author response to Decision Letter 1]

15 Jul 2025

Dear Editor,

We appreciate the comments and suggestions provided by the reviewers, and we have amended all the corrections and modifications accordingly.

Thank you very much.

---

## [Decision Letter · Decision Letter 1]

8 Aug 2025

Phytochemical and pharmacoinformatics analysis of a traditional antipsoriatic oil formulation for its potential against proinflammatory cytokines TNF-α and IL-17A

PONE-D-25-10185R1

Dear Dr. Chinta,

We’re pleased to inform you that your manuscript has been judged scientifically suitable for publication and will be formally accepted for publication once it meets all outstanding technical requirements.

Kind regards,

Yusuf Oloruntoyin Ayipo, Ph.D

Academic Editor

PLOS ONE

Additional Editor Comments (optional):

The study is timely and well-designed. Again, the submission meets the level of scientific rigour required for publication in this title, and all the concerns raised by the respective reviewers have been addressed satisfactorily. I hereby recommend the manuscript for publication in the current version.

Reviewers' comments:

Reviewer's Responses to Questions

**Comments to the Author**

Reviewer #2: All comments have been addressed

Reviewer #3: All comments have been addressed

2. Is the manuscript technically sound, and do the data support the conclusions?

Reviewer #2: Yes

Reviewer #3: Yes

3. Has the statistical analysis been performed appropriately and rigorously?

Reviewer #2: N/A

Reviewer #3: N/A

4. Have the authors made all data underlying the findings in their manuscript fully available?

Reviewer #2: Yes

Reviewer #3: Yes

5. Is the manuscript presented in an intelligible fashion and written in standard English?

Reviewer #2: Yes

Reviewer #3: Yes

Reviewer #2: Authors considered recommendations previously provided in the submitted revised manuscript, thus improving its quality and rigor.

Reviewer #3: The present study focuses on the phytochemical and pharmacoinformatics analysis of a traditional antipsoriatic oil formulation for its potential against proinflammatory cytokines TNF-α and IL-17A. The authors have addressed previous comments and the manuscript is now in a better shape.

**Do you want your identity to be public for this peer review?** For information about this choice, including consent withdrawal, please see our Privacy Policy

Reviewer #2: No

Reviewer #3: No

---

## [Editor Report · Acceptance letter]

PONE-D-25-10185R1

PLOS ONE

Dear Dr. Chinta,

I'm pleased to inform you that your manuscript has been deemed suitable for publication in PLOS ONE. Congratulations! Your manuscript is now being handed over to our production team.

Kind regards,

on behalf of

Dr. Yusuf Oloruntoyin Ayipo

Academic Editor

PLOS ONE